# Diagnostic Accuracy of Cone Beam Computed Tomography and Periapical Radiography for Detecting Apical Root Resorption in Retention Phase of Orthodontic Patients: A Cross-Sectional Study

**DOI:** 10.3390/jcm13051248

**Published:** 2024-02-22

**Authors:** Sónia A. Pereira, Ana Corte-Real, Ana Melo, Linda Magalhães, Nuno Lavado, João Miguel Santos

**Affiliations:** 1Institute of Orthodontics, Faculty of Medicine, University of Coimbra, 3000-075 Coimbra, Portugal; sapereira@fmed.uc.pt; 2Center for Innovation and Research in Oral Sciences (CIROS), Faculty of Medicine, University of Coimbra, 3000-075 Coimbra, Portugal; a.corte.real4@gmail.com; 3Forensic Dentistry Laboratory, Faculty of Medicine, University of Coimbra, 3000-075 Coimbra, Portugal; 4Dentistry Department, Faculty of Medicine, University of Coimbra, 3000-075 Coimbra, Portugal; acjmelo@gmail.com (A.M.); linda_guerreiro@hotmail.com (L.M.); 5Coimbra Institute of Engineering, Polytechnic Institute of Coimbra, 3030-199 Coimbra, Portugal; nlavado@isec.pt; 6Research Centre in Asset Management and System Engineering (RCM2+), Polytechnic Institute of Coimbra, 3030-199 Coimbra, Portugal; 7Institute of Endodontics, Faculty of Medicine, University of Coimbra, 3000-075 Coimbra, Portugal; 8Coimbra Institute for Clinical and Biomedical Research (iCBR) and Center of Investigation on Environment Genetics and Oncobiology (CIMAGO), Faculty of Medicine, University of Coimbra, 3000-548 Coimbra, Portugal

**Keywords:** external apical root resorption, periapical radiography, cone beam computed tomography, diagnostic accuracy, orthodontic treatment

## Abstract

**Objectives:** This clinical study aimed to evaluate and compare the diagnostic accuracy of intraoral periapical radiography (PR) and cone beam computed tomography (CBCT) in detecting external apical root resorption (EARR) in orthodontic patients during the retention phase. **Methods:** The research involved 41 Caucasian patients who had undergone comprehensive orthodontic treatment, with a total of 328 teeth analyzed. The Kappa values for inter- and intra-examiner agreement were high for both PR and CBCT, indicating a robust level of agreement among examiners. The study used a four-point scale for classifying EARR. **Results:** This study showed comparable accuracy, sensitivity, and specificity between PR and CBCT when using the most stringent criterion of “Definitely present”. The data suggested that CBCT outperformed PR when using a less stringent criterion (“Definitely present” or “Probably present”), particularly for maxillary incisors. However, overall diagnostic performance, as measured by the area under the ROC curve, showed only a slight advantage for CBCT over PR. Areas under the ROC curve range between 0.85 and 0.90 for PR and between 0.89 and 0.92 for CBCT. According to DeLong’s test, there is no evidence to conclude that the area under the ROC curve is different for PR and CBCT. **Conclusions:** Both PR and CBCT are accurate diagnostic tools for identifying EARR, with PR being deemed more suitable for routine clinical use due to its cost-effectiveness and lower radiation exposure. The findings emphasize the importance of considering the risk-benefit ratio when deciding on imaging modalities for monitoring EARR in orthodontic patients.

## 1. Introduction

External apical root resorption (EARR) is a common undesirable iatrogenic outcome of orthodontic treatment [1,2,3]. It is characterized by the permanent loss of apical radicular tissue (cementum and dentin) as a result of the action of clastic cells released during the inflammatory process induced by the orthodontic forces [4,5,6,7]. Root resorption is a complex phenotype with a multifactorial etiology associated with biological and mechanical factors [8,9,10,11]. Although all dentition can be affected by root resorption, the literature supports the idea that maxillary incisors are the most frequently and severely affected teeth, followed by mandibular incisors and first molars [12,13,14].

Much effort has been made to determine a scale to classify EARR severity. This is a hard task due to the complex etiology of root resorption and the difficulty of interpreting diagnostic exams. EARR is frequently classified as mild, moderate, severe, or extreme loss. In severe and extreme cases, root reduction can compromise tooth function and longevity [15,16,17]. Thus, it is extremely important to conduct radiographic monitoring of orthodontic patients to diagnose and control EARR. According to Fuss et al. [18], radiographically, the loss of root structure is located in the apical third of the root, and no signs of radiolucency can be observed in the bone or root. Several imaging exams, such as panoramic radiography, periapical radiography (PR), face profile teleradiography, and cone beam computed tomography (CBCT), have been used to diagnose EARR [19,20,21,22].

Periapical radiography is the most frequently used imaging method in daily practice [17]. The parallax technique allows us to obtain accurate and reproducible radiographic images and is used to define the lesion’s location [2,23,24]. However, being a two-dimensional technique, PR has limitations when evaluating the damage to the dental root. In addition, the overlap of anatomic structures and the distortion projection errors may result in blurred images [4,17,23,25]. Due to these factors, many studies claim that 3D lesions’ inference from PR is not reliable [4,26,27]. Furthermore, Andreasen et al. [28] and Goldberg et al. [29] reported that resorption lesions less than 0.3 mm deep and 0.6 mm diameter cannot be detected by periapical radiography.

CBCT has proven to be a promising alternative method to diagnose EARR [30]. This tool allows high-quality 3D image visualization of maxillofacial structures. Volumetric analysis of the root resorption provides accurate localization and quantification of the resorption lesions [25,27,31]. Compared with conventional computed tomography [2,5], CBCT needs lower doses of radiation and also produces images with no anatomical overlapping and with little distortion, allowing high sensitivity and specificity of images [5,25]. Conventional X-ray techniques, such as PR, require lower doses of radiation than CBCT, provided that the field of view (FOV) and voxel size are in the standard orthodontics clinical range [25]. Further, CBCT radiation doses can be multiplied by a factor of fifteen, depending on the selected resolution, for the same field of view [32].

Several studies have been published to discuss the ideal diagnostic exam (high validity and reliability) for root resorption. These studies used different samples and methods to obtain their results: teeth with external and internal inflammatory resorption (associated with endodontic treatment or not) [4,21]; artificial and natural root resorption in extracted teeth [17]; extracted deciduous teeth [23]. Freitas et al. [33] developed a pioneer in vivo study to assess differences in the frequency of EARR using 58 patients, explained by the imaging method (PR or CBCT). In the previous studies, the presence of EARR was only compared after orthodontic treatment related to long-term concerns. Evaluation PR versus CBCT as diagnostic tools were not yet reported in terms of accuracy, sensitivity, specificity, or other indicators, such as the Area Under the Receiver Operating Characteristic (ROC) Curve (AUC). The ROC curve shows the trade-off between sensitivity and specificity across different threshold settings for a classifier. It is a numerical measure of the ROC curve’s performance. The higher the AUC, the better the classifier is at distinguishing between positive and negative cases.

To the best of our knowledge, other than studies with impacted canines [34], there are no in vivo studies comparing the diagnostic accuracy of two imaging systems, PR and CBCT, for detecting EARR in orthodontic patients in the retention phase. The aim of this research was to evaluate and compare the accuracy of intraoral PR with CBCT in the diagnosis of external apical root resorption of orthodontic patients in the retention phase.

## 2. Materials and Methods

### 2.1. Data Collection

This study was a diagnostic accuracy cross-sectional study, and the patients from the Orthodontic Institute of the Faculty of Medicine of the University of Coimbra were invited to participate.

Those who accepted and met the inclusion criteria were selected, resulting in a study that included a total of 41 Caucasian patients.

The inclusion criteria for patient selection were established as follows: patients of Portuguese Caucasian origin; have performed a CBCT examination because of a particular condition of impacted wisdom teeth previously to this study; have received comprehensive orthodontic treatment (straight-wire technique); have a clinical file allowing the collection of the patient’s complete clinical information; patients in a retention phase for at least 6 months; with no genetic craniofacial malformation and no congenitally missing teeth; have no supernumerary or impacted canines or incisors; have no incisors with endodontic treatment and absence of periodontal disease.

The 41 patients included 14 males and 27 females, with an average age of 24.37 years (s.d. ± 5.9). In what concerns to the initial orthodontic diagnosis, all the patients had class I and Class II malocclusion, medium or thick gingival biotype, a reasonable quantity of bone in both incisor regions, and reasonable oral hygiene. In relation to the treatment plan, the average duration of orthodontic treatment was 27 ± 6 months; the patients didn’t need teeth extractions, and the incisor movements were retroinclination, proinclination, root torque, and intrusion. However, the intrusion was only in moderate amounts. All maxillary and mandibular incisors (eight teeth) were evaluated per patient, resulting in a total of 328 teeth being analyzed.

The purpose of this study was explained to all participants, and ethical committee approval was received from the Faculty of Medicine, University of Coimbra (ref. CE-020/2017, issued on 27 March 2017). Written consent forms were also obtained in accordance with the ethical principles of medical research and human rights, as stated in the Helsinki Declaration (2002 version, www.wma.net/e/policy/b3.htm, accessed on 3 January 2020), and details of such approval are included in the text.

### 2.2. Radiographic Technique

The digital periapical radiographs were taken by an intra-oral X-ray machine (Siemens^®^, Heliodent EC, 70 Dentotime, Munich, Germany) using a photosensitive phosphor plate (Carestream Dental, Atlanta, GA, USA, CS 7600, size 2—31 mm × 41 mm), with exposure parameters of 60 KV, 7 mA and 0.08 s. The images were obtained using the digital imaging system (Carestream Dental, CS 7600), and all the PR were obtained using the parallelism technique and an intraoral positioner Rinn XCP (Dentsply Sirona™, Charlotte, NC, USA). For each patient, three radiographs were performed (teeth 11 and 12; teeth 21 and 22; teeth 42, 41, 31, and 32), resulting in a total of 123 radiographs (Figure 1).

### 2.3. CBCT Technique

The DICOM images of CBCT were taken using 3D equipment (i-CAT^®^, Imaging Sciences International, Hatfield, PA, USA). Tomographic images were obtained with the following parameters: voxel size 0.25 mm, exposure time 14.5 s, and FOV (Field-of-View) 100–160 mm.

The CBCT slice analysis was performed with the OnDemand3D^TM^ App Software, https://www.ondemand3d.com assessed on 5 May 2021 (Cybermed, Inc., Seoul, Republic of Korea). All images were assessed on a portable computer in a dark room with no time limit. A video of the process is available as Appendix A. The 3D tool was selected, followed by the manual adjustment of the long axis of the anterior teeth in the upper and lower arch. The correct orientation of each tooth axis was performed in the three planes (Figure 2) with cut intervals of 0.125 mm and a thickness of 1 mm. For the dental analysis, the image of the sagittal plane coinciding with the corono-radicular longitudinal axis was chosen for each tooth (Figure 3).

### 2.4. Radiological Assessment

A committee of two skilled dental doctors with more than 20 years of clinical experience each decided, for each tooth, the presence or absence of EARR. This committee used PR and CBCT simultaneously to reach a consensus classification (presence or absence), the so-called gold-standard diagnostic for each tooth. Due to an inconclusive consensus diagnosis of mild root resorption, 104 teeth from the 41 patients were excluded from the study.

The remaining 224 teeth were evaluated by a second group of examiners, four post-graduate orthodontics students in their third year of training. Before making their judgment about EARR, these examiners were re-trained and reminded of the main characteristics of root resorption. A special training program was carried out with examples of EARR in CBCT and PR in order to solidify their knowledge of the imaging interpretation of the characteristics of the EARR. The examiners were only included in this study after having demonstrated the appropriate skills to evaluate accurately. Direct access to manipulate and view the 3D DICOM images themselves was given.

The selected examiners evaluated EARR using Patel et al. [5] modified scoring system with a four-point scale: 1—“Definitely present”; 2—“Probably present”; 3—“Probably absent”; 4—“Definitely absent” (Table 1). A sequential evaluation approach was established as follows: 1st session—analysis of PR, for a maximum of 5 patients per day; 2nd session—analysis of CBCT scans, for a maximum of 5 patients per day; 3rd session—evaluate/repeat PR and CBCT scans of 3 random patients (24 teeth for each exam). The purpose of the third session was to assess the intra-examiner agreement, which was performed two weeks after the previous sessions.

### 2.5. Data Analysis

The four-point scale used by examiners allows three different definitions of a positive test for EARR: (1) only if the interpretation was “Definitely present” (i.e., the most stringent criterion); (2) if the interpretation was “Definitely present” or “Probably present”; (3) if the interpretation was “Definitely present” or “Probably present” or “Probably not present” (i.e., the most lenient criterion). For the first two criteria, accuracy, sensitivity, specificity, and predictive values were determined using the gold standard of the consensus committee. These calculations were performed for each examiner and diagnostic test (PR and CBCT), and their means and standard deviations by diagnostic test were reported.

Three pairs of sensitivity and specificity values are needed to describe the overall performance of the diagnostic tests (CBCT and PR). For each examiner and diagnostic test, the area under the ROC curve, defined by those three dots, was used as a measure of overall diagnostic performance, and means and standard deviations by the diagnostic test were reported. Also, for each examiner, CBCT and PR areas under the ROC curve were non-parametrically compared using DeLong’s test.

Results were also reported separately for the following teeth groups: maxillary and mandibular incisors, mandibular incisors only, maxillary incisors only, central incisors, and lateral incisors.

A statistical computation was performed using R software (version 3.4.4) [34]. The R package *pROC* [35] was used to determine the area under the ROC curve and to perform DeLong’s test. Inter-examiner and intra-examiner agreements were assessed using the R package *irr* [36] and the metric ICC (Intra-class Correlation Coefficient), based on a mean rating and a 2-way random-effect model with a consistence agreement and absolute agreement, respectively.

## 3. Results

The Kappa value for inter-examiner agreement in the EARR diagnostic was 0.93 and 0.92 for PR and CBCT, respectively. The mean Kappa value for the intra-examiner agreement was 0.94 for both PR and CBCT.

Table 2 shows the frequency of teeth with EARR according to the consensus committee. EARR prevalence in this sample was between 67% (mandibular and central incisors) and 75% (lateral incisors).

Using the most stringent criterion for EARR presence (“Definitely present”), all the performance results are approximately the same for PR and CBCT. However, the data suggests that PR performance has more variability across examiners than CBCT (Table 3A). Specificity and positive predictive values are very high in all cases as a consequence of using the most stringent criterion. For both PR and CBCT, accuracy is quite low using these criteria, ranging from 0.49 (PR—mandibular incisors only) to 0.67 (CBCT—maxillary incisors only). EARR diagnosis in mandibular incisors turns out to be the hardest task for examiners.

Using the criteria for EARR presence “Definitely present” or “Probably present” (Table 3B), CBCT outperformed PR except for lateral incisors, where CBCT variability across examiners was higher than in other teeth groups. Overall, variability across examiners decreased when compared to the use of the “Definitely present” criterion only, and, as expected, specificity and positive predictive value have decreased. For both PR and CBCT, accuracy is good using these criteria, ranging from 0.77 (PR—maxillary incisors only and mandibular incisors only) to 0.93 (CBCT—maxillary incisors only).

Figure 4 depicts the ROC curve for each examiner for both PR and CBCT. Areas under the ROC curve are approximately the same for PR and CBCT. Thus, it suggests that, overall, using all three different definitions of a positive test for EARR, CBCT performs only slightly better than PR for all examiners.

However, according to DeLong’s test, there is no evidence to conclude that overall diagnostic performance is different for PR and CBCT (Table 4). It is worth noting that the results only suggest otherwise for one examiner and only for mandibular incisors. As the area under the ROC curve ranges between 0 and 1 and for both PR and CBCT values range between 0.83 (mandibular incisors) and 0.94 (maxillary and lateral incisors), it can be concluded that both have good ability to discriminate between teeth with and without EARR.

## 4. Discussion

During orthodontic treatment, radiological methods are crucial in identifying and monitoring root resorption due to the usual absence of clinical signs associated with this type of lesion before it reaches the latter stages [8,25,37].

Classification was performed using medical images (PR and CBCT) obtained after orthodontic treatment. The complexity of this task was increased by the fact that PR and CBCT images were not available before treatment. Consequently, the committee decided to exclude 104 out of the 328 teeth due to inconclusive consensus diagnoses about the presence of EARR. The excluded teeth were just those with mild EARR, thus out of the scope of clinical relevance.

A four-point scale was used by trained examiners to classify their level of confidence in detecting the presence of EARR, using PR and CBCT in different sessions. The Kappa values for inter-examiner agreement in the diagnosis of EARR were 0.93 and 0.94 for PR and CBCT, respectively, revealing a high level of agreement between examiners in this study. Patel et al. [5] had inter-examiner values of 0.365 and 0.925 for PR and CBCT, respectively, suggesting a poor concordance for PR. Schröder et al. [17] also reported an inter-examiner agreement greater than 0.80 for CBCT. The studies with high inter-examiner values are probably related to better training in PR interpretation. Regarding the intra-examiner agreement, Patel et al. [5] reported 0.625 for PR and 0.966 for CBCT, and Schröder et al. [17] obtained a mean value greater than 0.80 for CBCT. In the present study, the mean intra-examiner agreement was 0.94 for both PR and CBCT, suggesting a high level of agreement at different time points. The excellent concordance level for PR might be related to (1) the periapical radiography parallelism technique being well performed, (2) the use of digital PR instead of analogical PR, and (3) examiners who were well-trained in PR interpretation.

The scale used by the examiners in this study allowed three different definitions of a positive test for EARR. The strictest criterion, “Definitely present”, results in values for accuracy, sensitivity, specificity, positive predictive values, and negative predictive values for PR that differ from CBCT by less than 1%. Creanga et al. [38] and Schröder et al. [17] reported differences in sensitivity and specificity for PR and CBCT that range from 18% for sensitivity to 8% for specificity. Regarding the intermediate criterion for a positive test, “Definitely present” or “Probably present”, the accuracy of CBCT outperformed PR by about 6% for all incisors and about 16% for maxillary incisors. For lateral incisors, the accuracy of PR was better than CBCT by 6%. For both PR and CBCT, accuracy is higher using this criterion (77–93%) than the strictest criterion, “Definitely present”. EARR diagnosis of mandibular incisors turned out to be the hardest location for the examiners.

According to Metz [39], a value of 0.75–0.80 for the area under the ROC curve indicates that the imaging method is acceptable. In our study, the diagnostic accuracy for the detection of EARR was between 0.85 and 0.90 for PR and between 0.89 and 0.92 for CBCT. This suggests that CBCT performs only slightly better than PR. Therefore, based on areas under the ROC curve, we can conclude that both tests are accurate diagnostic tools to identify apical root resorption. The unexpectedly high performance of PR diagnostic could be a result of the irregular margins of root resorption lesions, which can be pathognomonic in this kind of lesion.

According to our study, PR and CBCT seem to be reliable tools to analyze and diagnose EARR. Our results are supported by a systematic review [25] concluding that the CBCT is a valuable tool to examine EARR during or at the end of orthodontic treatment. However, the average EARR measured with CBCT seems to lack clinical relevance. Another systematic review and meta-analysis [1] refer to a different idea, suggesting that CBCT has a higher diagnostic efficacy than PR. A recent study [40] of the diagnostic accuracy of CBCT and 2D imaging methods in the 3D localization and assessment of maxillary impacted canines suggests that the diagnostic accuracy of CBCT outperformed 2D radiography in localizing the position of the impacted canines and the resorption of the adjacent incisors. However, a comparison with the present study is difficult because they are treating a special situation of impacted canines.

In 2022, another recent systematic review [41] compared CBCT and panoramic radiography for the assessment of root resorption on the second molar associated with third molar impaction. This is relevant for our patients who present this condition. They concluded that more EARR is assessed in CBCT compared to panoramic, but there was considerable agreement between these two medical exams in the assessment of EARR, especially in the absence of the pathology rather than its presence.

Other authors [42] performed a comparative analysis of the accuracy of PR and CBCT for diagnosing complex endodontic pathoses, and CBCT had higher diagnostic accuracy in complex endodontic pathoses compared to PR. Nevertheless, CBCT failed to diagnose some pathologies in 33% of teeth, concluding that CBCT should be considered for selective cases where PR has diagnostic ambiguity.

Several authors have provided support for our study outcomes. Earlier investigations [8,26] found no significant disparities in the accuracy of root defect detection between periapical radiography (PR) and cone-beam computed tomography (CBCT). However, contrasting perspectives exist in the literature. Notably, Patel et al. [5] asserted that CBCT exhibits superior accuracy compared to PR in an endodontic clinical study. These findings align with the observations of Yi et al. [1], who demonstrated significantly elevated sensitivity and a larger area under the receiver operating characteristic (ROC) curve for CBCT based on a meta-analysis incorporating only five out of fifteen studies comparing PR to CBCT, all conducted in simulated models. It is crucial to acknowledge the limitations of such simulations, as induced lesions on the root surface, created artificially with mechanical instruments, may not faithfully replicate the true clinical nature of lesions. Given that resorption manifests with irregular cavities rather than perfect hemispherical shapes [2,4,6,17,38,43], the sensitivity and specificity of imaging methods may not accurately reproduce clinical conditions.

Scientific evidence [17] underscores the inherent challenge of observing and identifying in vivo external apical root resorption (EARR) compared to ex vivo scenarios, rendering a direct comparison with our in vivo results challenging. Furthermore, orthodontic research, relying on limited scientific evidence regarding performance metrics for the diagnosis and measurement of EARR, has been used to advocate the use of CBCT in some studies [44,45].

Our study presents some limitations: (1) lack of PR and CBCT images before orthodontic treatment (only used panoramic and cephalometric radiography) as PR and CBCT imaging methods were only used in an orthodontic retention phase, hampering the comparison of root length before and after treatment. However, this is not absolutely essential to the research once our aim was to evaluate and compare the accuracy of PR with CBCT in the diagnosis of EARR and not the development of root resorption due to the orthodontic treatment; (2) difficulty in applying an EARR quantitative measure and confirming the real extension of root resorption lesions (in vivo study); (3) the absence of a sample size calculation; and (4) the evaluation of the eight incisors instead of all the teeth.

The reason why we decided not to evaluate all teeth, and instead only the upper and lower incisors, was due to the high radiation that it would require to perform PR on all the teeth. Nevertheless, our work overcomes some limitations encountered in prior studies. First, in contrast to other in vivo investigations, this study encompasses a notably larger sample size [2,4,6,38,43]. Second, our approach avoids reliance on artificial lesions, enabling a genuine clinical assessment of imaging methods for root resorption. Consequently, our study is more pragmatic and rooted in everyday clinical observations. Lastly, our methodology facilitates performance analysis not only in terms of positive and negative outcomes for external apical root resorption (EARR) but also permits a nuanced clinical evaluation ranging from “Definitely present” to “Probably present.” These enhancements contribute to the robustness and clinical relevance of our findings, providing a more comprehensive understanding of the diagnostic capabilities of the assessed imaging methods in the context of EARR.

In clinical practice, PR is a reasonably priced, simple, and quick image acquisition tool [46]. However, for EARR detection, there are some limitations that require special care regarding the interpretation of the obtained results, namely the 2D images of a 3D structure, the presence of anatomical overlap, or the production of distorted and blurred images [37]. Due to these limitations, the detection and grading of smaller lesions situated in certain root surface sites become more difficult and less accurate [5,23]. Previous studies indicated that the 2D nature of the images of periapical radiography is less accurate in determining the location and severity/size of resorption lesions [15,38]. However, the acquisition of PR images from different angulations allows the gathering of more information. On the other hand, CBCT produces good-quality 3D images and eliminates anatomical overlaps [37], enabling the detection of EARR even when the lesions are smaller than 1 mm.

The present study did not find significant statistical differences in the performance of PR and CBCT for root resorption detection. Based on our findings, on the guidelines recently developed [19,21], on the high cost, on the radioactive burden, and based on ALARA (As Low as Reasonably Achievable) principles, CBCT should not be considered a routine exam, and its risk/benefit ratio must be taken into account.

Given that root resorption is an ongoing process necessitating vigilant monitoring throughout treatment, periapical radiography (PR) emerges as the preferred method for routine clinical use. However, in instances involving intricate cases or where there is a suspicion of a more severe form of external apical root resorption (EARR), cone-beam computed tomography (CBCT) can prove to be a pivotal asset [47]. CBCT facilitates a more comprehensive assessment, thereby enhancing informed decision-making for devising an optimal treatment plan and striving for the most favorable outcome.

In future clinical investigations, it would be worthwhile to undertake a similar study incorporating varying periapical radiographic angulations. Additionally, the inclusion of before-and-after treatment radiographic images could provide valuable insights, further advancing our understanding of the diagnostic capabilities of different imaging modalities in monitoring and assessing root resorption progression over the course of treatment.

## 5. Conclusions

This study contributes valuable insights by revealing the absence of significant differences in the accuracy of detecting external apical root resorption (EARR) associated with orthodontic treatment between periapical radiography (PR) and cone-beam tomography. In weighing the advantages and disadvantages of these two imaging methods, our findings suggest that periapical radiography (PR) stands out as the most suitable radiographic technique for the precise diagnosis of EARR in orthodontic clinical practice. This recommendation is grounded in the study’s rigorous evaluation of diagnostic accuracy and underscores the pragmatic utility of PR in effectively assessing and monitoring EARR in the orthodontic context.

## Figures and Tables

**Figure 1 jcm-13-01248-f001:**
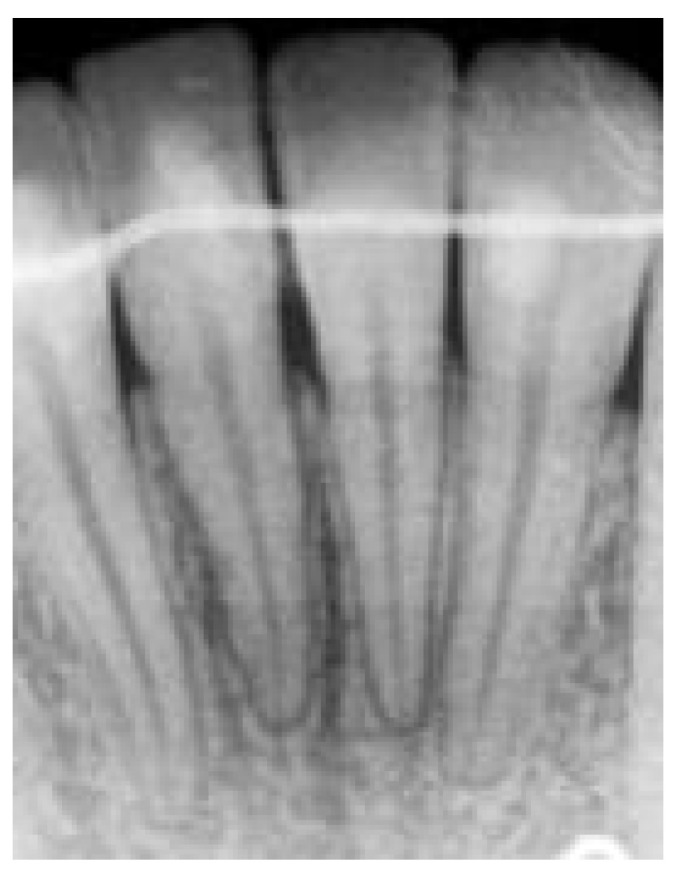
Sample of periapical radiography of the lower incisors assessed in this study.

**Figure 2 jcm-13-01248-f002:**
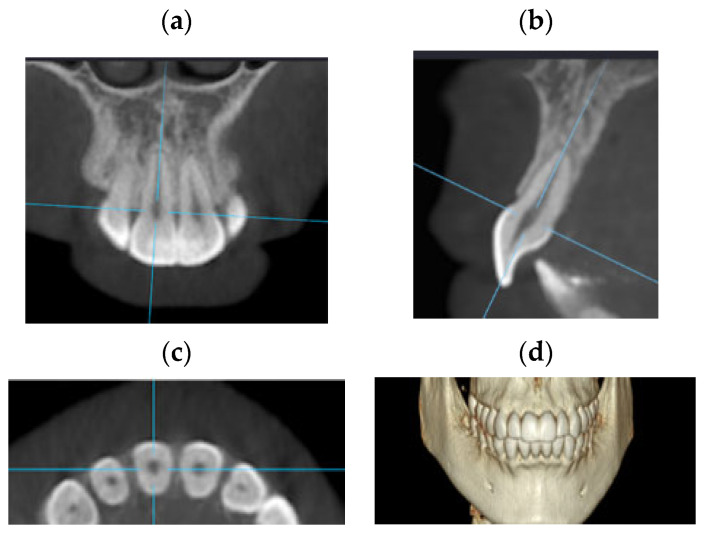
Example of 3-dimensional orientation for CBCT images. The correct orientation of each tooth axis was performed in the three planes: (**a**) coronal slice, (**b**) sagittal slice, (**c**) axial slice, and (**d**) 3D reconstruction.

**Figure 3 jcm-13-01248-f003:**
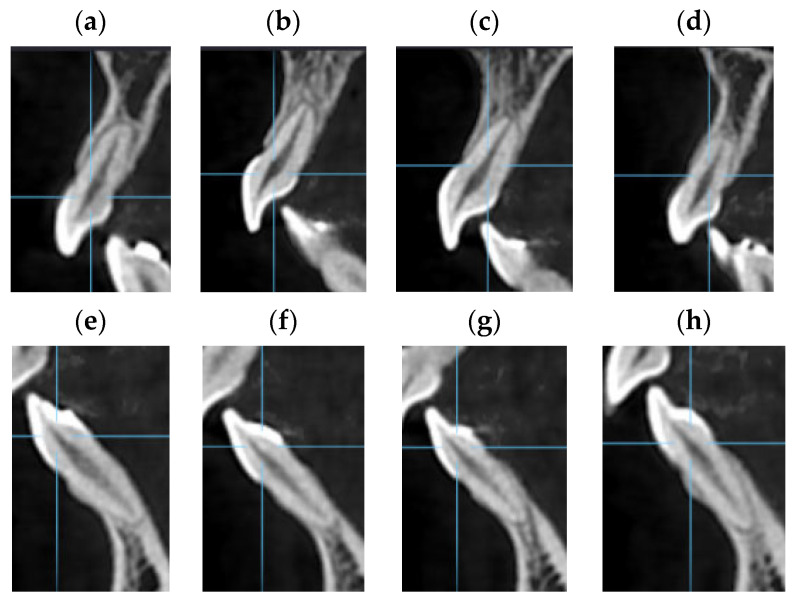
Example of CBCT Sagittal slices used to evaluate the upper and lower incisors of a patient, representing: (**a**) tooth 12; (**b**) tooth 11; (**c**) tooth 21; (**d**) tooth 22; (**e**) tooth 42; (**f**) tooth 41; (**g**) tooth 31; (**h**) tooth 32.

**Figure 4 jcm-13-01248-f004:**
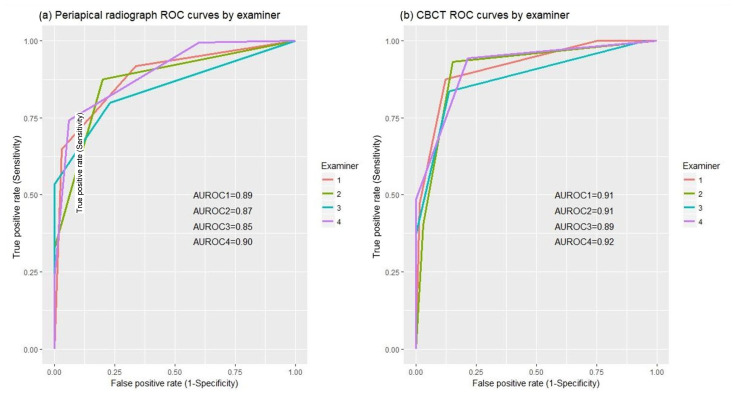
ROC curves by examiner: (**a**) using PR; (**b**) using CBCT. AUROC1, AUROC2, AUROC3, and AUROC4—non-parametric area under the ROC curve for examiner 1, 2, 3, 4, respectively.

**Table 1 jcm-13-01248-t001:** The questionnaire used to assess each tooth by the committee of two highly skilled dental doctors (Committee) and by the group of four post-graduate orthodontists (Examiners).

	Present		Absent	
Committee				
EARR	Definitely Present	Probably Present	Probably Absent	Definitely Absent
Examiners				

**Table 2 jcm-13-01248-t002:** Absolute frequency (relative frequency) of teeth with EARR according to the consensus committee.

Teeth Groups	EARR Present *n* (%)	EARR not Present *n* (%)	*n*
maxillary and mandibular incisors	159 (71%)	65 (29%)	224
maxillary incisors only	89 (74%)	31 (26%)	120
mandibular incisors only	70 (67%)	34 (33%)	104
central incisors	72 (67%)	36 (33%)	108
lateral incisors	87 (75%)	29 (25%)	116

**Table 3 jcm-13-01248-t003:** (**A**). Mean (standard deviation) of accuracy, sensitivity, specificity, PPV and NPV for EARR diagnostic using PR and CBCT at the most stringent criterion “Definitely present”. (**B**). Mean (standard deviation) of accuracy, sensitivity, specificity, PPV and NPV for EARR diagnostic using PR and CBCT with the criteria “Definitely present” or “Probably present”.

(**A**)						
Teeth groups		Accuracy	Sensitivity	Specificity	PPV	NPV
maxillary and mandibular incisors	PR	0.60 (0.13)	0.43 (0.19)	0.99 (0.02)	0.99 (0.09)	0.43 (0.08)
	CBCT	0.59 (0.04)	0.43 (0.05)	0.99 (0.01)	0.99 (0.01)	0.42 (0.02)
maxillary incisors only	PR	0.69 (0.14)	0.59 (0.20)	0.98 (0.03)	0.99 (0.01)	0.48 (0.12)
	CBCT	0.67 (0.06)	0.56 (0.07)	0.98 (0.03)	0.99 (0.02)	0.44 (0.05)
mandibular incisors only	PR	0.49 (0.13)	0.24 (0.19)	1 (0.00)	1 (0.00)	0.40 (0.07)
	CBCT	0.50 (0.05)	0.26 (0.08)	0.99 (0.01)	0.99 (0.03)	0.49 (0.03)
central incisors	PR	0.63 (0.14)	0.45 (0.22)	0.99 (0.03)	0.99 (0.02)	0.49 (0.1)
	CBCT	0.64 (0.03)	0.47 (0.04)	0.99 (0.03)	0.99 (0.03)	0.48 (0.02)
lateral incisors	PR	0.56 (0.13)	0.42 (0.17)	1 (0.00)	1 (0.00)	0.37 (0.07)
	CBCT	0.55 (0.05)	0.40 (0.07)	0.99 (0.02)	0.99 (0.01)	0.36 (0.03)
(**B**)						
Teeth groups		Accuracy	Sensitivity	Specificity	PPV	NPV
maxillary and mandibular incisors	PR	0.82 (0.03)	0.83 (0.08)	0.79 (0.11)	0.91 (0.04)	0.67 (0.08)
	CBCT	0.88 (0.03)	0.89 (0.05)	0.84 (0.04)	0.93 (0.01)	0.78 (0.08)
maxillary incisors only	PR	0.77 (0.17)	0.79 (0.29)	0.73 (0.23)	0.91 (0.07)	0.69 (0.25)
	CBCT	0.93 (0.01)	0.96 (0.03)	0.85 (0.09)	0.95 (0.03)	0.87 (0.06)
mandibular incisors only	PR	0.77 (0.05)	0.73 (0.12)	0.87 (0.10)	0.93 (0.05)	0.62 (0.08)
	CBCT	0.83 (0.07)	0.82 (0.10)	0.84 (0.09)	0.91 (0.04)	0.71 (0.11)
central incisors	PR	0.79 (0.02)	0.84 (0.07)	0.71 (0.13)	0.86 (0.05)	0.70 (0.07)
	CBCT	0.88 (0.02)	0.92 (0.03)	0.82 (0.03)	0.91 (0.02)	0.84 (0.05)
lateral incisors	PR	0.84 (0.04)	0.83 (0.08)	0.90 (0.10)	0.96 (0.03)	0.65 (0.10)
	CBCT	0.78 (0.17)	0.74 (0.26)	0.90 (0.10)	0.96 (0.03)	0.60 (0.20)

PPV—positive predictive value NPV—negative predictive value.

**Table 4 jcm-13-01248-t004:** Mean (standard deviation) of the area under the ROC curve for PR and CBCT.

Teeth Groups	PR	CBCT	*p*-Value ^a^
maxillary and mandibular incisors	0.87 (0.02)	0.90 (0.01)	0.294, 0.228, 0.199, 0.436
maxillary incisors only	0.91 (0.01)	0.94 (0.03)	0.072, 0.523, 0.131, 0.361
mandibular incisors only	0.83 (0.02)	0.86 (0.05)	0.747, 0.012, 0.671, 0.803
central incisors	0.85 (0.02)	0.91 (0.02)	0.074, 0.281, 0.151, 0.189
lateral incisors	0.91 (0.02)	0.91 (0.02)	0.745, 0.430, 0.729, 0.476

^a^ DeLong’s test for the area under the ROC curve comparison. *p*-value for each examiner.

## Data Availability

The data presented in this study are available on request from the first and the last authors.

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
