# Peer review of "Diagnostic Accuracy of Cone Beam Computed Tomography and Periapical Radiography for Detecting Apical Root Resorption in Retention Phase of Orthodontic Patients: A Cross-Sectional Study"

_jcm, 2024, doi:10.3390/jcm13051248_

Round 1
Reviewer 1 Report
Comments and Suggestions for Authors
1- I think it will be more reliable if the committee consisted of dental radiologists instead of just skilled dentists.
2-Regarding the CBCT voxel size it has been reported that the proper voxel size for EARR detection ranged from 0.150 mm3 up-to 0.250 mm3. The chosen size may not detect or accurately detect EARR.
3- References have old dates and need to be updated.
Comments on the Quality of English LanguageThe writing language in this piece is moderately effective. While it generally conveys the intended message, there are areas where clarity could be improved. Sentence structures are varied, contributing to a decent flow, but attention to smoother transitions between ideas would enhance the overall coherence. Additionally, considering the audience and adjusting the level of technicality might improve accessibility for a broader readership. Overall, a moderate quality of writing language with potential for refinement in terms of clarity and audience engagement.
Author Response
1-I think it will be more reliable if the committee consisted of dental radiologists instead of just skilled dentists.
AU: Thanks for your valuable comment. In our country, we have radiologists or neuroradiologists, but we do not have dental radiologists. In our field, orthodontics, an Orthodontic specialist or a 3rd Year Postgraduate has a substantial experience in viewing and interpretation of CBCT and periapical radiographs. In this way, we consider they have the best ability and training to perform the assessment, and they actually will be the professionals that need to do that evaluation in the clinical setting of our country.
2-Regarding the CBCT voxel size it has been reported that the proper voxel size for EARR detection ranged from 0.150 mm3 up-to 0.250 mm3. The chosen size may not detect or accurately detect EARR.
AU: Thanks for your valuable comment. This was an error during the writing process. We double-checked and corrected the parameters description in METHODS section …“Tomographic images were obtained with the following parameters: voxel size 0.25 mm, exposure time 14.5 seconds and FOV (Field-of-View) 100-160mm.”
3-References have old dates and need to be updated.
AU: Thanks for your valuable comment. We have updated the references with the following papers:
Alfailany DT, Shaweesh AI, Hajeer MY, Brad B, Alhaffar JB. The diagnostic accuracy of cone-beam computed tomography and two-dimensional imaging methods in the 3D localization and assessment of maxillary impacted canines compared to the gold standard in-vivo readings: A cross-sectional study. Int Orthod 2023;21:100780. https://doi.org/10.1016/J.ORTHO.2023.100780.
Moreira-Souza L, Butini Oliveira L, Gaêta-Araujo H, Almeida-Marques M, Asprino L, Oenning AC. Comparison of CBCT and panoramic radiography for the assessment of bone loss and root resorption on the second molar associated with third molar impaction: a systematic review. Dentomaxillofac Radiol 2022;51:20210217. https://doi.org/10.1259/dmfr.20210217.
Keerthana G, Singh N, Yadav R, Duhan J, Tewari S, Gupta A, et al. Comparative analysis of the accuracy of periapical radiog-raphy and cone-beam computed tomography for diagnosing complex endodontic pathoses using a gold standard reference – A prospective clinical study. Int Endod J 2021;54:1448–61. https://doi.org/10.1111/iej.13535.
Reviewer 2 Report
Comments and Suggestions for Authors
Thanks for inviting me to review this manuscript.
I have the following points that require some attention from the authors. I would be grateful to hear good responses regarding the raised points.
Title
1- The title should be improved to encompass the teeth being studied (upper and lower incisors), the type of patients being treated (orthodontic patients in the retention phase), and the study design (i.e., a cross-sectional study).
Abstract
2- Some numbers (i.e., statistics) should be inserted in the "Results" subheading. This Abstract appeared free of any mean or median values. We must see some numbers, p-values, and AUCs under the ROCs. Sensitivity, specificity, and diagnostic accuracy values should be inserted anywhere.
Introduction
3- Line 93: The authors claim that no published study compares the 2D and 3D methods of assessing root resorption in orthodontic patients. This claim is inaccurate; they should consult and read this paper: Alfailany DT, Shaweesh AI, Hajeer MY, Brad B, Alhaffar JB. The diagnostic accuracy of cone-beam computed tomography and two-dimensional imaging methods in the 3D localization and assessment of maxillary impacted canines compared to the gold standard in-vivo readings: A cross-sectional study. Int Orthod. 2023;21(3):100780. doi:10.1016/j.ortho.2023.100780. This paper includes an assessment of root resorption of the adjacent teeth (i.e., incisors) by comparing 2D versus 3D assessments. This paper should be cited, and the results of the current findings should be compared with those.
Materials and Methods
4- Line 106: What is the reason for obtaining a CBCT image for these patients? What is the rationale for subjecting these patients to unnecessary radiation? There are very specific conditions for taking a CBCT image. Could you explain the reasons beyond taking these shots?
5- LIne 156: How can we consider the consensus of two doctors to be the gold standard? The gold standard should be set by seeing the defect (when extracting the tooth being resorbed) or by artificially creating this defect. The consensus of two doctors cannot be considered the gold standard. Both doctors may not arrive at the correct diagnosis of the root lesion. Why did you not use a panel of five or six doctors to make your gold standard more robust? The gold standard in this study is a problem.
6- Line 159: One-third of the teeth were excluded from the study due to the inconclusiveness of the diagnosis. This is a great proportion! This may affect the validity of the obtained results from this study. We need to know more about these 33% of the sample. These 33% of the sample may confuse the observers when using 2D and 3D methods. The exclusion of one-third of the sample makes the results of this study biased.
7- Line 159: Why did you choose only four postgraduate students? Usually, assessment panels are based on 8-10 persons or more. Using only four 3rd-year postgraduate students is not enough to give robust results.
8- Line 168: The four-point Likert scale is faulty. What is the difference between Point 2 and Point 3? What is the difference between "probably absent" and "probably present."? Points 2 and 3 are the same. Therefore, I think that the scale is not good for this study. I was expecting the assessors to rate the severity of resorption into (1) nothing, (2) mild, (3) moderate, and (4) severe. Unfortunately, this was not done.
9- Table 1 is not needed, and it should be deleted. It is a repetition of what was mentioned in the text.
Results
10- Figure 4: the term on the Y-axis should be "sensitivity" instead of "sensibility." in the two ROC curves.
Discussion
11- The authors should compare their results with studies that are in direct relation to this study.
12- The limitation section should be improved to reflect all the previously raised points regarding (1) the absence of a true gold standard in this study, (2) the small number of persons in the consensus committee, (3) the shortcomings with the panel of assessors, (4) the four-point scale that was employed in the current work, (5) the need to evaluate all teeth instead on only the upper an lower incisors, (6) the absence a sample size calculation.
Comments on the Quality of English LanguageIt should be improved.
Author Response
Thanks for inviting me to review this manuscript.
I have the following points that require some attention from the authors. I would be grateful to hear good responses regarding the raised points.
Title
- The title should be improved to encompass the teeth being studied (upper and lower incisors), the type of patients being treated (orthodontic patients in the retention phase), and the study design (i.e., a cross-sectional study).
AU: Thanks for your valuable comment. We agree with the suggestion of giving a more precise title to the manuscript and changed to: “Diagnostic accuracy of Cone Beam Computed Tomography and Periapical Radiography for Detecting Apical Root Resorption in Retention phase of orthodontic patients: A cross-sectional study”. Including more information in the title do not seem adequate, because it is already very long.
Abstract
- Some numbers (i.e., statistics) should be inserted in the "Results" subheading. This Abstract appeared free of any mean or median values. We must see some numbers, p-values, and AUCs under the ROCs. Sensitivity, specificity, and diagnostic accuracy values should be inserted anywhere.
AU: We agree, some statistics have been added in the abstract. Nevertheless, details about sensitivity, specificity, and diagnostic accuracy values are dependent on the criteria for EARR presence and thus not easy to summarize. Abstract has been changed to include: “Areas under the ROC curve ranges between 0.85 and 0.90 for PR and between 0.89 and 0.92 for CBCT. According to DeLong’s test, there is no evidence to conclude that the area under the ROC curve is different for PR and CBCT.”
Introduction
- Line 93: The authors claim that no published study compares the 2D and 3D methods of assessing root resorption in orthodontic patients. This claim is inaccurate; they should consult and read this paper:Alfailany DT, Shaweesh AI, Hajeer MY, Brad B, Alhaffar JB. The diagnostic accuracy of cone-beam computed tomography and two-dimensional imaging methods in the 3D localization and assessment of maxillary impacted canines compared to the gold standard in-vivo readings: A cross-sectional study. Int Orthod. 2023;21(3):100780. doi:10.1016/j.ortho.2023.100780. This paper includes an assessment of root resorption of the adjacent teeth (i.e., incisors) by comparing 2D versus 3D assessments. This paper should be cited, and the results of the current findings should be compared with those.
AU: Thanks for your valuable comment. We changed the sentence on line 93. “To the best of our knowledge, other than studies with impacted canines (ref. 40), there are no in vivostudies comparing the diagnostic accuracy of two imaging systems, PR and CBCT, for detecting EARR in orthodontic patients in retention phase”. The suggested paper has been cited in the introduction and compared with ours in the discussion.
Materials and Methods
4-Line 106: What is the reason for obtaining a CBCT image for these patients? What is the rationale for subjecting these patients to unnecessary radiation? There are very specific conditions for taking a CBCT image. Could you explain the reasons beyond taking these shots?
AU: Thanks for your valuable comment. We do not use the CBCT as a part of a routine protocol, since we follow the principle of ALARA. All of these patients, had the particular condition of impacted wisdom teeth, in the orthodontic retention phase. This was the particular reason why we planned the CBCT, reason included in the manuscript …”because of a particular condition of impacted wisdom teeth, needing extraction”.
5-LIne 156: How can we consider the consensus of two doctors to be the gold standard? The gold standard should be set by seeing the defect (when extracting the tooth being resorbed) or by artificially creating this defect. The consensus of two doctors cannot be considered the gold standard. Both doctors may not arrive at the correct diagnosis of the root lesion. Why did you not use a panel of five or six doctors to make your gold standard more robust? The gold standard in this study is a problem.
AU: The committee of two skilled dental doctors had 28 years of clinical experience. This is the best and possible committee for pragmatical reasons related to the design of this study, since we cannot perform histological analysis without damaging the teeth and submit the patients to unnecessary invasive clinical procedures. This Gold standard has been used by many authors previously, for example by Patel et al in some studies (ref. 5). We used a panel of two doctors, following the tendency in this kind of study.
6-Line 159: One-third of the teeth were excluded from the study due to the inconclusiveness of the diagnosis. This is a great proportion! This may affect the validity of the obtained results from this study. We need to know more about these 33% of the sample. These 33% of the sample may confuse the observers when using 2D and 3D methods. The exclusion of one-third of the sample makes the results of this study biased.
AU: Thanks for your valuable comment. We used a sample of 41 patients, the big sample that we found, in similar works. We assessed many teeth (328) and due to inconclusive consensus diagnosis (mild root resorption), 104 teeth, were excluded from the study. However, 224 teeth, remained in the study. We believe that besides this, we still had a big number of teeth to assess and this fact does not compromise the results. Added text: “due to mild root resorption”(line 161)
7-Line 159: Why did you choose only four postgraduate students? Usually, assessment panels are based on 8-10 persons or more. Using only four 3rd-year postgraduate students is not enough to give robust results.
AU: The postgraduation in our University has the duration of 3 years, but in full-time and exclusivity. The students have classes 5 days a week, during all the day and they don´t work outside. We enroll 4 students per year and we expect that by the end of the 3rth year, they have good skills, in diagnosis, planning and treatment. For this reason, we did not want to include students with less education time (in lower years of the postgraduation).
8- Line 168: The four-point Likert scalis faulty. What is the difference between Point 2 and Point 3? What is the difference between "probably absent" and "probably present."? Points 2 and 3 are the same. Therefore, I think that the scale is not good for this study. I was expecting the assessors to rate the severity of resorption into (1) nothing, (2) mild, (3) moderate, and (4) severe. Unfortunately, this was not done.
AU: We agree that the four-point ordinal scale could have different labels. Our choice was based in previous publications in this field of research, particularly the paper [ref. 5] and we decided to maintain the same experimental protocol for the sake of results’ comparison.
9- Table 1 is not needed, and it should be deleted. It is a repetition of what was mentioned in the text.
AU: We think that table 1 helps the reader. The other 4 reviewers were neutral about it, so we decided to keep it.
.10- Figure 4: the term on the Y-axis should be "sensitivity" instead of "sensibility." in the two ROC curves.
AU: Thanks for your pertinent comment. The Y-Axis label was updated accordingly.
Discussion
11- The authors should compare their results with studies that are in direct relation to this study.
AU: Thank you for the suggestion. We already had many references of authors with studies related to ours in the discussion. However, we added recent studies directly related to our study (line 297-312).
12- The limitation section should be improved to reflect all the previously raised points regarding (1) the absence of a true gold standard in this study, (2) the small number of persons in the consensus committee, (3) the shortcomings with the panel of assessors, (4) the four-point scale that was employed in the current work, (5) the need to evaluate all teeth instead on only the upper an lower incisors, (6) the absence a sample size calculation.
AU: The limitations were already mentioned but we increased the points mentioned (line 339-348).
Reviewer 3 Report
Comments and Suggestions for Authors
Please calculate and report the power of the study based on your sample size.
Please update your references so you have more recent references in the manuscript. Several references appear outdated; consider incorporating more recent literature to strengthen the theoretical framework.
Some sentences are complex and may impede understanding. Simplifying language without compromising precision would enhance clarity.
Comments on the Quality of English Language
The manuscript will benefit from a language revision.
Author Response
Please calculate and report the power of the study based on your sample size.
AU: No, we didn’t perform sample size calculation before patient’s selection. Notice that, the statistical test (DeLong’s test) is related to the significant difference between the AUC from CBCT and AUC from PR and it is a nonparametric test. In section about the study limitations (line 328) we now included this: (4) the absence a sample size calculation.
Please update your references so you have more recent references in the manuscript. Several references appear outdated; consider incorporating more recent literature to strengthen the theoretical framework.
AU: Thanks for your valuable comment. We have updated the list of references, as previously explained to reviewer 1.
Some sentences are complex and may impede understanding. Simplifying language without compromising precision would enhance clarity.
AU: Thanks for your valuable comment. We have polished the manuscript.
Reviewer 4 Report
Comments and Suggestions for Authors
Dear Author
Thank you for your huge sound work. I wish if you write some words about the sample size calculation and whether the average duration of fixed orthodontic treatment and whether there was any cases treated by intrusion of anterior teeth or not.
Regards
Author Response
Dear Author
Thank you for your huge sound work. I wish if you write some words about the sample size calculation and whether the average duration of fixed orthodontic treatment and whether there was any cases treated by intrusion of anterior teeth or not.
AU: Thanks for your valuable comment. As explained to reviewer 3 “No, we didn’t perform sample size calculation before patient’s selection. Notice that, the statistical test (DeLong’s test) is related to the significant difference between the AUC from CBCT and AUC from PR and it is a nonparametric test. In section about the study limitations (line 328) we now included this: (4) the absence a sample size calculation.”
The average duration of fixed orthodontic treatment was 26+/-7 months. The orthodontic treatments involved some intrusion of the anterior teeth, in the most part of the cases, in order to correct the overbite and overjet to ideal values. None of the patients were subjected to teeth extraction.
Reviewer 5 Report
Comments and Suggestions for Authors
This is a very interesting study through which the authors aim to assess a critical matter in orthodontic clinical practice. However, there are several things that need to be improved in order for this study to be appropriate for publicaation.
1. Since only incisors were examined, the title should be properly modified in order to clearly define this important detail.
2. There is an important lack of information regarding the orthodontic treatment details of the assessed patients, such as initial orthodontic diagnosis (exact type of malocclusion, inclination of lower incisors, gingival biotype, oral hygiene of the patients, bone quality and quantity in the incisor region) and the treatment plan that was followed (potential extractions, treatment duration, performed incisor root position corrections). If these are not reported, there is a high risk of bias in grouping unsimilar initial baseline data and thus reporting false conclusions.
3. The reasons why the CBCTs were deemed appropriate for these patients should be explained. Was there a particular detailed that was planned to be investigated via the CBCT or is CBCT a part of a routine protocol that is implemented in the respective setting?
4. How was the root resorption defined? Were there initial and final radiographic records that were compared? What were the exact points/planes of reference that were used for the comparison?
5. Was a sample size calculation performed before the selection of the patients? If yes, what was the variable used for this selection and what was considered as "minimal clinically important effect"?
6. A flow diagram depicting the process for the patient selection should be included
7. The detection of root resorption itself, even though it is an important diagnostic finding, yet it should be further assessed in matters of the exact extent that this resoprtion was observed, in order to provide a more clinically evident conclusion. For example, it would be interesting to find if there is a difference in detecting root resorption between the two assessed radiograhic techniques from the early stages (1-2 mm of resorption) or from more advanced stages (for example 4 mm or more).
8. The exclusion of these 104 teeth from the assessment should be better justified.
Author Response
This is a very interesting study through which the authors aim to assess a critical matter in orthodontic clinical practice. However, there are several things that need to be improved in order for this study to be appropriate for publication.
- Since only incisors were examined, the title should be properly modified in order to clearly define this important detail.
AU: Thanks for your valuable comment. Reviewer 1 already made some comments about that issue, and our answer is: We agree with the suggestion of giving a more precise title to the manuscript and changed to: “Diagnostic accuracy of Cone Beam Computed Tomography and Periapical Radiography for Detecting Apical Root Resorption in Retention phase of orthodontic patients: A cross-sectional study”. Including more information in the title do not seem adequate, because it is already very long.
- There is an important lack of information regarding the orthodontic treatment details of the assessed patients, such as initial orthodontic diagnosis (exact type of malocclusion, inclination of lower incisors, gingival biotype, oral hygiene of the patients, bone quality and quantity in the incisor region) and the treatment plan that was followed (potential extractions, treatment duration, performed incisor root position corrections). If these are not reported, there is a high risk of bias in grouping unsimilar initial baseline data and thus reporting false conclusions.
A: Thanks for your valuable comment. The studied sample has already been characterized with the relevant information for the variables assessed in this research. Nevertheless, we included some additional information as requested.
Added in line 117-122 …“In what concerns to the initial orthodontic diagnosis, all the patients had class I and Class II malocclusion, medium or thick gingival biotype, reasonable quantity of bone in both incisor region and reasonable oral hygiene. In relation to the treatment plan: the average duration of orthodontic treatment was of 27+/-6 months; the patients didn´t need teeth extractions, the incisor movements were retroinclination, proinclination, root torque and intrusion. However, the intrusion was only in moderate amounts.”
- The reasons why the CBCTs were deemed appropriate for these patients should be explained. Was there a particular detailed that was planned to be investigated via the CBCT or is CBCT a part of a routine protocol that is implemented in the respective setting?
AU: Thanks for your valuable comment. As previously explained to reviewer 2 “We do not use the CBCT as a part of a routine protocol, since we follow the principle of ALARA. All of these patients, had the particular condition of impacted wisdom teeth, in the orthodontic retention phase. This was the particular reason why we planned the CBCT, reason included in the manuscript …” because of a particular condition of impacted wisdom teeth, needing extraction”.
4-How was the root resorption defined? Were there initial and final radiographic records that were compared? What were the exact points/planes of that were used for the comparison?
A: Thanks for your valuable comment. The root resorption was defined, radiographically, as a loss of root structure, in the apical third of the root (blunting or shortening of the root apex) and no signs of radiolucency can be observed in the bone or root.
There are initial and final digital panoramic and cephalometric radiographies, as a routine protocol, to diagnosis and treatment. We only have final periapical radiography and CBCT, in the retention phase.
About the points/planes used on periapical radiography and CBCT for evaluation and EARR detection in the retention phase, we used the long axis of the anterior teeth, in the upper and lower arch, but the focus was around the apical third of each root.
When we used the CBCT (a video of the process is available as supplementary material) we performed a manual adjustment of the long axis of the anterior teeth, in the upper and lower arch. The correct orientation of each tooth axis was performed in the three planes: coronal, sagittal and axial. We also did the analysis specially around the apical third of each root.
5-Was a sample size calculation performed before the selection of the patients? If yes, what was the variable used for this selection and what was considered as "minimal clinically important effect"?
AU: Thanks for your valuable comment. As previously explained to reviewers 3 and 4, “No, we didn’t perform sample size calculation before patient’s selection. Notice that, the statistical test (DeLong’s test) is related to the significant difference between the AUC from CBCT and AUC from PR and it is a nonparametric test. In section about the study limitations (line 328) we now included this: (4) the absence a sample size calculation.”
6-A flow diagram depicting the process for the patient selection should be included
AU: Thanks for your valuable comment. The patient selection has been explained in the manuscript with the details, at materials and methods, from line 107 until line 128.
7-The detection of root resorption itself, even though it is an important diagnostic finding, yet it should be further assessed in matters of the exact extent that this resoprtion was observed, in order to provide a more clinically evident conclusion. For example, it would be interesting to find if there is a difference in detecting root resorption between the two assessed radiograhic techniques from the early stages (1-2 mm of resorption) or from more advanced stages (for example 4 mm or more).
AU: This is a very pertinent question, thank you. Indeed, we found that there is a difference in detecting root resorption between the two assessed radiographic techniques. When the resorption is of mild severity, the CBCT is the ideal technique to identify it. However, in these cases the root resorption does not have an important clinical effect. We found that when the root resorption is of moderate or severe loss (more than 5 mm), the periapical radiography, when well performed is good enough to detect it.
8- The exclusion of these 104 teeth from the assessment should be better justified.
AU: Thanks for your comment. We used a sample of 41 patients, the big sample that we found, in similar works. We assessed many teeth (328) and due to inconclusive consensus diagnosis (mild root resorption), 104 teeth, were excluded from the study. However, 224 teeth, remained in the study. We believe that besides this, we still have a big number of teeth to analyze. We decided to assess a lower number of teeth, but with the best Gold standard evaluation. In the cases where the resorption was of mild severity, the CBCT was the ideal technique to identify it, but it was difficult to do an exact evaluation to create the gold standard. Added: due to mild root resorption (line 161).
Round 2
Reviewer 2 Report
Comments and Suggestions for Authors
The authors have managed to address the majority of my raised points.
No need for further changes since the study design cannot be altered.
Comments on the Quality of English LanguageSome minor editing is still required.